# Cord Blood Manganese Concentrations in Relation to Birth Outcomes and Childhood Physical Growth: A Prospective Birth Cohort Study

**DOI:** 10.3390/nu13124304

**Published:** 2021-11-28

**Authors:** Yiming Dai, Jiming Zhang, Xiaojuan Qi, Zheng Wang, Minglan Zheng, Ping Liu, Shuai Jiang, Jianqiu Guo, Chunhua Wu, Zhijun Zhou

**Affiliations:** 1Key Laboratory of Public Health Safety of Ministry of Education, Key Laboratory of Health Technology Assessment of National Health Commission, School of Public Health, Fudan University, No.130 Dong’an Road, Shanghai 200032, China; 20111020031@fudan.edu.cn (Y.D.); zhangjiming@fudan.edu.cn (J.Z.); xjqi@cdc.zj.cn (X.Q.); 20211020149@fudan.edu.cn (Z.W.); mlzheng@cdc.zj.cn (M.Z.); 0557084@fudan.edu.cn (P.L.); jiangshuai12@fudan.edu.cn (S.J.); jqguo14@fudan.edu.cn (J.G.); chwu@fudan.edu.cn (C.W.); 2Zhejiang Provincial Center for Disease Control and Prevention, No.3399 Binsheng Road, Hangzhou 310051, China

**Keywords:** manganese, cord blood, birth outcomes, childhood growth

## Abstract

Gestational exposure to manganese (Mn), an essential trace element, is associated with fetal and childhood physical growth. However, it is unclear which period of growth is more significantly affected by prenatal Mn exposure. The current study was conducted to assess the associations of umbilical cord-blood Mn levels with birth outcomes and childhood continuous physical development. The umbilical cord-blood Mn concentrations of 1179 mother–infant pairs in the Sheyang mini birth cohort were measured by graphite furnace atomic absorption spectrometry (GFAAS). The association of cord-blood Mn concentrations with birth outcomes, and the BMI z-score at 1, 2, 3, 6, 7 and 8 years old, were estimated separately using generalized linear models. The relationship between prenatal Mn exposure and BMI z-score trajectory was assessed with generalized estimating equation models. The median of cord-blood Mn concentration was 29.25 μg/L. Significantly positive associations were observed between Mn exposure and ponderal index (β, regression coefficient = 0.065, 95% CI, confidence interval: 0.021, 0.109; *p* = 0.004). Mn exposure was negatively associated with the BMI z-score of children aged 1, 2, and 3 years (β = −0.383 to −0.249, *p* < 0.05), while no significant relationships were found between Mn exposure and the BMI z-score of children at the age of 6, 7, and 8 years. Prenatal Mn exposure was related to the childhood BMI z-score trajectory (β = −0.218, 95% CI: −0.416, −0.021; *p* = 0.030). These results indicated that prenatal Mn exposure was positively related to the ponderal index (PI), and negatively related to physical growth in childhood, which seemed most significant at an early stage.

## 1. Introduction

Manganese (Mn), an abundant heavy metal on Earth that occurs naturally [1], is widely used in industry, including the manufacturing of cosmetics, fertilizer, paints, fireworks, and the additive agents in gasoline and pesticides [2,3]. At the same time, Mn is an essential trace element and functions as a cofactor in critical biological processes that are involved in bone formation and the metabolism of carbohydrates, amino acids, and lipids [4,5]. However, excess Mn intake or exposure is associated with adverse neurological outcomes in children [6]. One review pointed out that Mn supplementation of infant formulas and excess Mn intake from drinking water should be avoided due to the potential hazards of excess Mn [7]. Animal studies have related both prenatal Mn deficiency and overexposure to decreased fetal size [8,9,10]. The primary route of typical Mn intake in humans is through diet and drinking water, but an additional amount of Mn can also enter the human body through the respiratory system and via skin contact from the environment [11]. Generally, 2 mg of Mn per day constitutes an adequate intake for pregnant women, with a tolerable upper intake level of 11 mg, according to the Food and Nutrition Board of the Institute of Medicine [12].

The fetus is particularly susceptible to environmental threats. Gestational exposure to environmental toxicants has been related not only to fetal growth and development but also to long-term health challenges [13,14,15]. Thus, assessing the relationship between prenatal pollutants exposure and the features of childhood growth is desirable, and identifying which period of growth is most sensitive to prenatal environmental pollutants exposure is essential for infants’ and children’s development. Mn can be transferred from the mother to the fetus via the placenta through active transport mechanisms [16]. Thus, the concentration of Mn in cord blood is able to clearly present the level of exposure to Mn of the fetus [17].

Since the required amount of Mn can be taken in through dietary sources, Mn deficiency is rare in humans. However, excess Mn exposure or Mn deficiency during pregnancy has been associated with fetal development. Several epidemiologic studies have revealed inconsistent relationships between Mn exposure during pregnancy and birth size [10,18,19]. For instance, a study in Wuhan in China has suggested that high levels of urinary Mn, even at levels that did not exceed the upper limit of reference (10 μg/L), are related to an increased risk of low birth weight [20]. However, data from Spain suggested that placental Mn was associated with slight increases in head circumference [21]. Nevertheless, limited evidence showed prenatal exposure to Mn was associated with continuous childhood anthropometric measures. Based on a longitudinal birth cohort in Jiangsu Province, China, this study aimed to assess the association of prenatal Mn exposure with birth size as well as childhood physical growth, and to explore which period of children’s growth is most sensitive to prenatal Mn exposure.

## 2. Materials and Methods

### 2.1. Study Population

The Sheyang mini birth cohort study (SMBCS) is a prospective birth cohort study of fetal and childhood exposure to environmental chemicals and their impact on growth and neurodevelopment in children [22]. In total, 1303 pregnant women were recruited at a major maternity hospital between June 2009 and January 2010 in Sheyang County, Jiangsu Province, China [23,24,25]. Pregnant women who volunteered to participate in the study signed an informed consent form and agreed to donate cord blood samples. After the exclusion of 15 pregnant women with complications (14 with gestational diabetes mellitus and 1 with hyperthyroidism), 4 active smokers and 7 who consumed alcohol, 1 still-birth, 9 congenital anomalies, 9 multiple births, 52 cases with incomplete information and 22 showing a lack of Mn concentrations in cord blood, eventually, 1179 mother–newborn pairs were eligible for analyzing birth outcomes. Subsequently, the present study focused on mother–child pairs that had at least one indicator of child anthropometry at ages 1, 2, 3, 6, 7 or 8 years (Appendix A). The recruited women provided written informed consents and each child’s caregivers also signed for their child. The SMBCS protocol was approved by the Ethics Committee of the School of Public Health, Fudan University (IRB#2021−02−0875).

### 2.2. Umbilical-Cord Blood Mn Analysis

Umbilical cord blood samples were collected by professional midwives, using standard protocols [26]. Whole blood samples were collected in 5 mL sterile centrifuge tubes containing anticoagulant EDTA and were stored at −80 °C until analysis. Mn concentrations in umbilical-cord blood were measured using graphite furnace atomic absorption spectrometry (GFAAS, Perkin Elmer AA 800, Waltham, MA, USA), as described previously [27]. Briefly, samples were mixed with 1:9 (*v*/*v*) of 0.1% nitric acid (guaranteed reagent (GR)). The external quality-control program did not show any time trend in the accuracy of the Mn measurement. The recovery of cord blood Mn was 94.0% ~ 102.7% and relative standard deviation (RSD) was 1.0% ~ 5.2%. The limit of detection (LOD) was 0.062 μg/L.

### 2.3. Anthropometric Measurements

Anthropometric measurements of the newborns, including their weight, length and head circumference, were assessed by hospital staff [28]. Birth weight was measured using a digital scale and was rounded to the nearest 0.1 kg. Birth length was measured with extended legs and heels against the measuring board, using an infantometer, and rounded to the nearest 0.1 cm. Head circumference was measured to the closest 0.1 cm at the maximal occipital-frontal circumference, using a standard measuring tape. The ponderal index (PI), known to be a good indicator used to quantify asymmetric fetal growth restriction and to reflect adiposity in infants, was calculated using mass divided by the height, cubed (PI = weight(g)/height(cm)^3^ × 100) [29].

The children’s weight and length were also measured when assessments were followed up at ages 1, 2, 3, 6, 7 and 8 years. In each visit, we measured the weight to the nearest 0.1 kg and height to the closest 0.1 cm. Body mass index (BMI) was calculated as the child’s weight in kilograms divided by the square of their height in meters. We computed age- and sex-standardized z-scores using the World Health Organization (WHO)’s child growth standards. Therefore, the final measures of body sizes were z-scores for weight, height, head circumference, weight for height and BMI.

### 2.4. Covariates

Medical history, such as gestational age and gestational weight gain, was abstracted from medical records. A questionnaire was administered to each pregnant woman upon recruitment, to collect information on socio-demographic characteristics, living environment and lifestyles. The data on the child’s health and behavior were collected with specifically designed questionnaires by trained study staff during the study’s follow-up visits. Potential confounding variables were adjusted, including the known or suspected risk factors for the exposure or outcome a priori, based on the previous literature and a directed acyclic graph (DAG) (Appendix A). In the multivariable regression models, a set of potential confounders were included if they were related to Mn exposure and body size (*p* < 0.10) or changed the coefficients of Mn concentrations by more than 10% (Appendix A). The following covariates for an analysis of birth outcomes were included: maternal age at delivery, pre-pregnancy BMI, gestational age, gestational weight gain, maternal education (<high school or ≥high school), family annual income (<30,000 RMB or ≥30,000 RMB), parity (0 or ≥1), passive smoking (yes or no), vitamin supplement during pregnancy (yes or no), child’s sex and child’s birth weight. The delivery mode was also adjusted in models for head circumference to correct head shape after spontaneous vaginal deliveries. The child’s age in months and their time spent playing outdoors (<3 h or ≥3 h) were additionally included to investigate prenatal Mn exposure and childhood anthropometric parameters.

### 2.5. Statistical Analysis

Cord blood Mn concentrations were naturally logarithmically transformed to normalize their skewed distributions. Independent *t*-tests for continuous variables and Chi-square tests for categorical variables were used to assess potential differences in characteristics between initial mother–newborn pairs and included participants at each period. Missing values in the covariables were imputed by multiple imputations (less than 5%).

Generalized linear models were applied to evaluate the associations between Mn concentration and the anthropometric measurements of the children at each age stage. Separate models were fitted for Mn exposure and for each outcome. We further stratified the models according to the children’s sex. Moreover, we assessed the linearity of each dose-response relationship between Mn exposure and each outcome, using generalized additive models (GAMs) with a three-degrees-of-freedom cubic spline function. Because no non-linear exposure-response relationship was observed, linear models were still used. Associations between prenatal Mn exposure and longitudinal BMI z-score were examined using multivariable generalized estimating equation models (GEE).

We performed GAM model analysis using SAS (version 9.4, SAS Institute Inc., Cary, NC, USA) and the other analyses in SPPS version 19.0. Statistical significance was considered as a two-sided *p*-value < 0.05.

### 2.6. Sensitivity Analysis

We performed several sensitivity analyses. Heavy metals could impact fetal and child growth; we further adjusted the lead and cadmium concentrations in cord blood to test the robustness of the results. We also restricted the participants, excluding low birth-weight infants and preterm births. Furthermore, we re-ran the multivariable generalized estimating equation models using the data of children who were followed up at all time points. 

## 3. Results

### 3.1. General Characteristics

Table 1 presents mother–child pairs’ characteristics from pregnancy until the child is 8 years of age. In total, 1072 (90.9%) pregnant women were younger than 35 years old at delivery. The gestational period of 1170 (99.2%) infants was over 37 weeks. Before conception, 149 (12.6%) women were underweight (BMI < 18.5 kg/m^2^), 842 (71.4%) were normal weight (BMI between 18.5 and 23.9 kg/m^2^), and 188 (16.0%) were overweight and obese (BMI ≥ 24 kg/m^2^). A majority of women (63.9%) had less than a high-school education. In addition, approximately 52.5% of the newborns were boys. The status of socioeconomic information at childhood was similar to that during pregnancy; no significant difference was observed regarding the distributions of baseline characteristics among enrollments and in six follow-up visits.

### 3.2. Mn Concentrations in Cord Blood

Cord blood Mn levels were detectable in all samples. The median level of Mn in cord blood was 29.25 μg/L, ranging from 6.84 μg/L to 316.73 μg/L (Appendix A). The geometric mean and geometric standard deviation of cord-blood Mn concentrations were 29.03 μg/L and 1.50 μg/L. 

### 3.3. Cord-Blood Mn Concentration and Birth Outcomes

Associations between Mn exposure and size at birth were listed in Table 2. After controlling for potential confounders, each 1-unit increase in ln-transformed Mn concentrations in cord blood was associated with an increase of 0.065 (95% CI: 0.021, 0.109) in PI of newborns at birth (*p* = 0.004), about 2.51% of the mean PI (mean: 2.59). Prenatal exposure to Mn was associated with an increase in PI in the fourth quartiles of exposure compared to the lowest quartile (β = 0.082, 95% CI: 0.032, 0.132; *p* = 0.001). In addition, the infants among the highest quartile of prenatal Mn exposure had a higher birth weight compared to the lowest quartile (β = 0.067, 95% CI: 0.003, 0.131; *p* = 0.041).

In sex-stratified analyses, higher Mn exposure was in association with increases in PI among girls (β = 0.079, 95% CI: 0.011, 0.146; *p* = 0.022). In addition, among girls, prenatal Mn concentrations were positively related to birth weight (β = 0.090, 95% CI: 0.010, 0.170; *p* = 0.027). The dose-response relationships between prenatal Mn exposure and birth outcomes were shown in Figure 1. Generalized additive models suggested a linear relationship between cord blood Mn and PI among the total infants, and the same linear association between cord blood Mn and birth weight among girls.

### 3.4. Cord-Blood Mn Level and BMI of Children Aged 1, 2, 3, 6, 7, and 8 Years Old

As shown in Figure 2, prenatal Mn exposure was negatively associated with the BMI z-score of children at the ages of 1 year (β = −0.383, 95% CI: −0.668, −0.098; *p* = 0.008), 2 years (β = −0.300, 95% CI: −0.546, −0.055; *p* = 0.017), and 3 years (β = −0.249, 95% CI: −0.477, −0.020; *p* = 0.033). However, non-significantly inverse associations of prenatal Mn exposure with childhood BMI z-score at the ages of 6, 7 and 8 years were observed.

Regarding sex-specific difference (Appendix A), Mn concentrations in cord blood were significantly inversely related to the BMI z-score in boys aged 1 year old (β = −0.607, 95% CI: −0.993, −0.220; *p* = 0.002) and 3 years old (β = −0.358, 95% CI: −0.710, −0.005; *p* = 0.047). A significant negative relationship between cord-blood Mn concentration and BMI z-score was observed in 2-year-old girls (β = −0.434, 95% CI: −0.805, −0.062; *p* = 0.022).

In longitudinal analyses using GEE models (Figure 3, Appendix A), prenatal Mn exposure was related to the childhood BMI z-score trajectory in children aged 1 to 8 years old (β = −0.253, 95% CI: −0.445, −0.060; *p* = 0.010). A significant trend was also observed from the first quartile to the fourth quartile (*p* = 0.016). Compared with the lowest Mn levels in cord blood, the BMI z-score was significantly decreased in the fourth quartile (β = −0.234, 95% CI: −0.460, −0.009; *p* = 0.041). These inverse associations between Mn exposure and childhood BMI trajectory were only observed in boys (β = −0.388, 95% CI: −0.678, −0.098; *p* = 0.009).

### 3.5. Sensitivity Analysis

Overall, effect estimates did not change in terms of complete analyses when adding other heavy metals, such as lead and cadmium (Appendix A). Results were similar after excluding those children who were low-birth-weight infants and preterm births (Appendix A). In addition, the data of children followed up at all time points made the coefficients become stronger (Appendix A). Finally, the results remained robust after performing sensitivity analysis.

## 4. Discussion

In this prospective birth cohort, we found significant associations of prenatal Mn exposure with birth size and physical development during childhood. Briefly, prenatal Mn exposure was associated with an increased PI among newborns. Higher Mn concentrations in cord blood were related to BMI during toddlerhood, but not in the school-age period. The association of Mn with the childhood BMI z-score trajectory feature was also observed.

The level of cord blood Mn in the present study (median = 29.25 μg/L, geometric mean = 29.03 μg/L, range: 6.84–316.73 μg/L) was within the range of levels seen in prior studies. Our measured level was comparable to the median value of Mn concentration, as measured in Germany (median = 28.8 μg/L) [30] and Canada (median = 31.8 μg/L) [31] but was much lower than the levels reported in the US, Mexico, and Beijing in China. (>40 μg/L, Appendix A) [6,17,32]. The differences in cord-blood Mn concentrations across populations could be a result of environmental exposure levels and dietary intakes. Blood Mn has been advised as a biomarker of Mn exposure [33,34]. Cord-blood Mn represents fetal exposure during the third trimester because the half-life of Mn in the blood is 37 days [35]. However, further studies with sequential measures of Mn during pregnancy are needed to confirm our findings and explore whether the associations of Mn with birth outcomes and childhood physical development depend on the timing of Mn exposure.

Only a few epidemiologic research studies addressing cord-blood Mn and birth size with inconsistent results have been conducted [8,19,36,37]. Specifically, in a study of 1377 mother-infant pairs in Shanghai, China, the Mn concentrations in cord serum were significantly related to shorter birth length and higher PI [38]. In agreement with their findings, we observed positive associations between Mn in cord blood and the PI of infants. Similarly, our findings showed that cord-blood Mn concentrations were related to a reduction in birth length, but the association was not significant. At the same time, we found higher levels of cord-blood Mn were associated with increased birth weight compared with those showing the lowest Mn levels. Another Chinese study in northern China using data from 125 mother–infant pairs showed an inverse U-shaped relationship between cord-blood Mn levels (median = 77.20 μg/L) and birth weight [19]. In Spain, cord-blood Mn was not significantly associated with birth weight, probably due to the small sample size (*n* = 54) [8]. A study of 1519 mother–infant pairs in Canada found a negative association for cord-blood Mn with birth weight [36]. The level of exposure might be a reason for these inconsistent results of the associations between cord-blood Mn and birth weight. An animal study also suggested that prenatal exposure to Mn was related to reduced fetal body weight in mice [37]. Although some negative associations between cord-blood Mn and birth weight were found, the potential mechanisms might include oxidative stress caused by excess Mn exposure [39,40]. Mn is still an essential trace element that was shown to play a role in bone formation and fetal growth. The level of Mn concentrations in our study captured the ascending segment of the inverted U-shape dose-response relationship between Mn and birth weight. In addition, a study from Bangladesh suggested sex-specific positive and linear associations for Mn with birth weight [41]. Our sex-stratified results were similar. Potential mechanisms for the associations are unclear; prenatal sex-steroid hormones [42] and maternal iron deficiency [43] might play an important role.

As for postnatal growth, we found that prenatal Mn exposure was negatively associated with childhood BMI z-score, and the association was more significant when the toddlers were aged 1, 2, or 3 years old, but non-significant at the age of 6, 7, or 8 years. Human growth and development at different stages were influenced by several kinds of factors that include micronutrients (e.g., Mn, etc.) [44,45]. A cross-sectional study analyzing two cycles of NHANES revealed that the increase in the concentration of blood Mn was associated with an increased BMI in US children aged 6–12 years [46]. Another study of 470 preschool children on the southeast coast of China indicated that there was no significant association between children’s blood Mn and BMI [15], in contrast to the present study. Mn exposure in utero or in childhood might have different effects on children’s growth. According to the Development Origins of Health and Disease (DOHaD), prenatal exposure to environmental factors played a role in determining the development of human growth in childhood [47]. Associations between prenatal Mn and BMI in the development periods of children were found in the present study and were also found in the sensitivity analysis of data from the children followed up at all time points. However, which periods of childhood are mostly affected by prenatal Mn exposure was unclear. To the best of our knowledge, this is the first study that illuminated the associations between prenatal Mn exposure and BMI z-score at the ages of 1, 2, 3, 6, 7, or 8 years. We also found that prenatal Mn exposure was more sensitive in terms of BMI z-score in toddlerhood, and the associations became weaker in school-age childhood. Our further study is to explore longer-term effects, even physical development, in adolescence.

Benefiting from the continuous follow-up of our cohort, we could observe the physical growth of children from birth to the age of 8. Prenatal Mn exposure was associated with an increased PI at birth and decreased BMI z-score in childhood, especially toddlerhood. Catch-up, which is defined as accelerated rates of growth following a period of failure to reach the growth reference of normal preterm or term-born infants, seemed to explain the results [48]. As an essential element, moderate Mn intake in utero could ensure the normal growth of the fetus, prevent low birth weight, and avoid catch-up. Furthermore, Mn exposure was suspected to affect human health in a “U-shaped” dose-responsive manner [10,18]. Excess Mn exposure or Mn deficiency might have different mechanisms. The potential biological mechanisms caused by prenatal excess Mn exposure could be related to impairments in fetal development and childhood growth, but the processes were still unclear. The animal studies suggested that maternal excessive Mn exposure was related to reduced fetal weight, the restricted internal organ development of offspring, and impaired skeleton ossification [37,49,50]. Conversely, Mn deficiency also resulted in skeletal malformation and impaired growth in animals [4,51]. Moreover, high levels of Mn exposure were seen to cause oxidative stress in cells and impair the function and growth of cells [39]. Besides this, Mn deficiency was associated with impairments or dysfunctions in insulin production, lipoprotein metabolism, the oxidant defense system, and growth factor metabolism [52,53]. Therefore, more studies are needed to explore these unclear mechanisms.

Obviously, our study had several strengths and limitations. The repeated anthropometric data from a prospective cohort that had been followed for more than 8 years provided us with a good opportunity to explore the associations of cord-blood Mn with physical growth in utero and in toddlerhood, up to school-age childhood, although the research showed a lack of data for children at the ages of 4 and 5 years due to there being no follow-up. Reassuringly, participants who enrolled in the study could be representative of the full cohort and the presented results were still robust in the sensitivity analysis. On the other hand, potential confounders were obtained from face-to-face interviews, and recall bias was avoided. Although we adjusted the potential confounders as soon as possible, some other important determinants, such as some other microelements (e.g., Cu, Fe, Zn) [54], could not be controlled, which might bias the results. Therefore, future studies on the mixture effect of different microelements and other heavy metals on birth outcomes and childhood growth are needed.

## 5. Conclusions

In this prospective birth cohort study, we found that a high level of cord-blood Mn was associated with a significant increase in birth weight and ponderal index. Prenatal exposure to Mn was inversely related to childhood BMI z-score, especially in toddlerhood at the age of 1, 2, and 3 years. Further studies are warranted to confirm our findings and explore a longer-term effect in adolescence.

## Figures and Tables

**Figure 1 nutrients-13-04304-f001:**
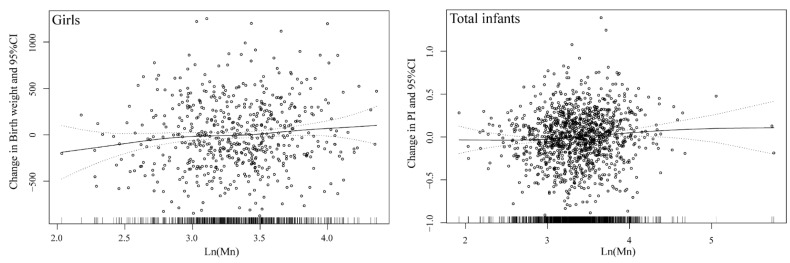
Generalized additive models for associations between prenatal Mn exposure and birth outcomes.

**Figure 2 nutrients-13-04304-f002:**
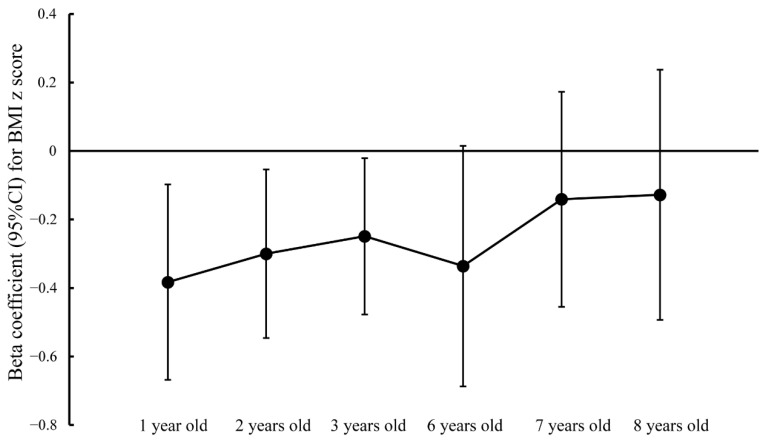
Generalized linear models for associations between prenatal Mn exposure and body mass index z score at different ages. (Abbreviation: BMI—body mass index).

**Figure 3 nutrients-13-04304-f003:**
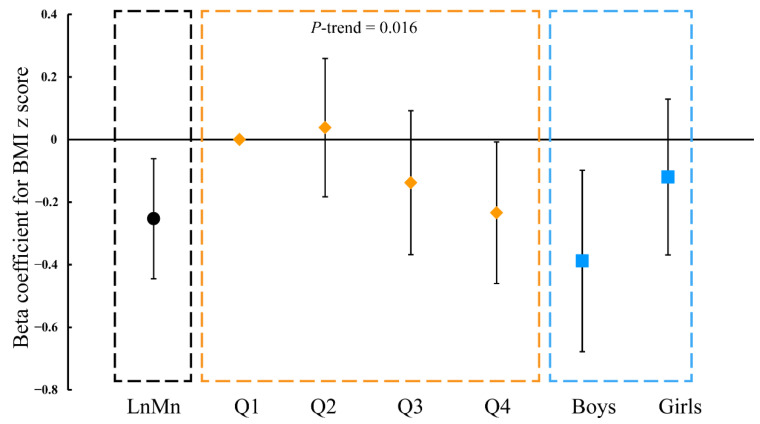
Generalized estimating equation models for associations of body mass. index z score with prenatal Mn exposure. (Abbreviation: BMI—body mass index).

**Table 1 nutrients-13-04304-t001:** Maternal characteristics and sociodemographic characteristics in each subgroup. (*N* (%)).

Characteristics	Pregnancy (*n* = 1179)	1 Year Old (*n* = 567)	2 Years Old (*n* = 358)	3 Years Old (*n* = 409)	6 Years Old (*n* = 421)	7 Years Old (*n* = 388)	8 Years Old (*n* = 374)	*p* *
Maternal age (years)								
<35	1072 (90.9)	507 (89.4)	323 (90.2)	374 (91.4)	370 (87.9)	344 (88.7)	332 (88.8)	
≥35	107 (9.1)	60 (10.6)	35 (9.8)	35 (8.6)	51 (12.1)	44 (11.3)	42 (11.2)	0.461
Gestational age (weeks)								
<37	9 (0.8)	3 (0.5)	2 (0.6)	2 (0.5)	4 (1.0)	3 (0.8)	2 (0.5)	
≥37	1170 (99.2)	564 (99.5)	356 (99.4)	407 (99.5)	417 (99.0)	385 (99.2)	372 (99.5)	0.975
Pre-pregnancy BMI (kg/m^2^)								
<18.5	149 (12.6)	74 (13.1)	52 (14.5)	53 (12.9)	43 (10.2)	41 (10.6)	42 (11.2)	
18.5–23.9	842 (71.4)	405 (71.4)	248 (69.3)	287 (70.2)	294 (69.8)	269 (69.3)	254 (67.9)	
≥24	188 (16.0)	88 (15.5)	58 (16.2)	69 (16.9)	84 (20.0)	78 (20.1)	78 (20.9)	0.288
Maternal education								
<High school (9 years)	753 (63.9)	388 (68.4)	252 (70.4)	284 (69.4)	302 (71.7)	272 (70.1)	272 (72.7)	
≥High school (9 years)	426 (36.1)	179 (31.6)	106 (29.6)	125 (30.6)	119 (28.3)	116 (29.9)	102 (27.3)	0.006
Neonatal sex								
Boys	619 (52.5)	296 (52.2)	194 (54.2)	202 (49.4)	227 (53.9)	216 (55.7)	203 (54.3)	
Girls	560 (47.5)	271 (47.8)	164 (45.8)	207 (50.6)	194 (46.1)	172 (44.3)	171 (45.7)	0.663
Parity								
0	613 (52.0)	293 (51.7)	196 (54.7)	224 (54.8)	225 (53.4)	212 (54.6)	192 (51.3)	
≥1	566 (48.0)	274 (48.3)	162 (45.3)	185 (45.2)	196 (46.4)	176 (45.4)	182 (48.7)	0.848

* The differences in frequency distribution were established by chi-square tests. Abbreviation: BMI: body mass index.

**Table 2 nutrients-13-04304-t002:** Regression coefficients (95% CI) for associations between Mn concentrations in cord blood and birth outcomes.

	Birth Weight (kg)		Birth Length (cm)		Head Circumference (cm)		Ponderal Index	
	β (95% CI)	*p*	β (95% CI)	*p*	β (95% CI)	*p*	β (95% CI)	*p*
All newborns ^a^								
Ln (Mn)	0.044 (−0.012, 0.100)	0.124	−0.190 (−0.508, 0.128)	0.242	0.105 (−0.093, 0.303)	0.299	0.065 (0.021, 0.109)	0.004
Q1	Ref.		Ref.		Ref.		Ref.	
Q2	0.033 (−0.031, 0.096)	0.316	0.059 (−0.301, 0.420)	0.748	0.097 (−0.125, 0.320)	0.391	0.032 (−0.018, 0.081)	0.209
Q3	0.035 (−0.029, 0.098)	0.283	−0.031 (−0.391, 0.329)	0.865	0.012 (−0.211, 0.236)	0.914	0.042 (−0.008, 0.091)	0.101
Q4	0.067 (0.003, 0.131)	0.041	−0.160 (−0.522, 0.202)	0.385	0.193 (−0.031, 0.418)	0.091	0.082 (0.032, 0.132)	0.001
*p*-trend		0.051		0.331		0.172		0.002
Boys ^b^								
Ln (Mn)	0.005 (−0.073, 0.083)	0.899	−0.378 (−0.804, 0.047)	0.082	0.104 (−0.183, 0.392)	0.477	0.052 (−0.005, 0.110)	0.073
Q1	Ref.		Ref.		Ref.		Ref.	
Q2	0.050 (−0.047, 0.146)	0.312	0.143 (−0.381, 0.667)	0.592	0.173 (−0.178, 0.524)	0.334	0.023 (−0.048, 0.093)	0.528
Q3	0.056 (−0.039, 0.151)	0.245	0.194 (0.323, 0.710)	0.462	0.090 (−0.258, 0.438)	0.611	0.002 (−0.068, 0.071)	0.962
Q4	0.054 (−0.039, 0.147)	0.256	−0.287 (−0.792, 0.219)	0.266	0.285 (−0.056, 0.626)	0.102	0.078 (0.010, 0.146)	0.024
*p*-trend		0.288		0.260		0.159		0.043
Girls ^b^								
Ln (Mn)	0.090 (0.010, 0.170)	0.027	0.043 (−0.435, 0.521)	0.861	0.109 (−0.158, 0.376)	0.424	0.079 (0.011, 0.146)	0.022
Q1	Ref.		Ref.		Ref.		Ref.	
Q2	0.012 (−0.072, 0.095)	0.784	−0.035 (−0.531, 0.461)	0.889	0.027 (−0.248, 0.302)	0.848	0.035 (−0.035, 0.976)	0.323
Q3	0.011 (−0.073, 0.096)	0.792	−0.281 (−0.782, 0.221)	0.273	−0.038 (−0.317, 0.241)	0.788	0.081 (0.011, 0.152)	0.024
Q4	0.085 (−0.003, 0.172)	0.059	0.019 (−0.503, 0.542)	0.943	0.096 (−0.193, 0.385)	0.516	0.080 (0.006, 0.153)	0.034
*p*-trend		0.089		0.755		0.665		0.013

^a^ Models were adjusted for maternal age at delivery, pre-pregnancy BMI, gestational age, gestational weight gain, maternal education, parity, family annual income, passive smoking, vitamin supplement during pregnancy, child’s sex, delivery mode (just for head circumference). ^b^ Models were adjusted for maternal age at delivery, pre-pregnancy BMI, gestational age, gestational weight gain, maternal education, parity, family annual income, passive smoking, vitamin supplement during pregnancy, delivery mode (just for head circumference).

## Data Availability

The data presented in this study are available on request from the corresponding author. The data are not publicly available due to privacy.

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
