# Peer review of "Cord Blood Manganese Concentrations in Relation to Birth Outcomes and Childhood Physical Growth: A Prospective Birth Cohort Study"

_nutrients, 2021, doi:10.3390/nu13124304_

Round 1
Reviewer 1 Report
Review
Nutrients
Cord blood manganese concentrations in relation to birth outcomes and childhood physical growth: a prospective birth co-ort study
Yiming Daia,1, Jiming Zhanga,1, Xiaojuan Qia,b, Zheng Wanga, Minglan Zhenga, Ping Liua, Shuai Jianga, Jianqiu Guoa, Chunhua Wua and Zhijun Zhoua.
Dai et al present a prospective epidemiological study conducted over 8 years to better understand the effect of in utero Mn exposure on growth and during childhood. This work is very interesting because the study is well done and involves a large number of children, more than 1100. As the authors point out, manganese is an essential element but is toxic in excess and is associated with negative effects on growth and neurodevelopment. Although there is a consensus on the toxicity of manganese, these effects remain difficult to measure as shown by the sometimes contradictory results of epidemiological studies. The results of Dai et al. are therefore important because they provide new information to better understand these effects.
Please find my comments and suggestion on the manuscript:
Line 24 : « 2.9.25µg/L. »
What is the correct value ? (see 29.3 μg/L line 169)
Line 25 : « β=0.065. »
At first reading, I thought that the β was a correlation coefficient (Pearson or Spearman) but it is not, it’s the regression coefficient of the linear relationship between Ln(Mn concentration) and ponderal index (if I understood correctly). This should be clearly stated in the abstract (idem for all β values).
Line 25, 26 : « Mn exposure was negatively associated with body mass index z score of children aged 1, 2, and 3 years (β=-0.375~-0.244, p<0.05) was observed… »
Two verbs, please make corrections. What means « ~ » ? is it the range of β values ? in this case replace by « to ».
Line 42 : « through diet »
Suggestion « through diet and drinking water »
Line 46. This is a hot topic with a world public health dimension. It would be interesting to add in the manuscript that the World Health Organization (WHO) has recently published proposed revision to its drinking water guideline for manganese. The revision is designed specifically to protect infants from excess manganese in infant formula and drinking water. (https://www.who.int/docs/default-source/wash-documents/wash-chemicals/gdwq-manganese-background-document-for-public-review.pdf). A recent paper should be added here in reference :
How much manganese is safe for infants? A review of the scientific basis of intake guidelines and regulations relevant to the manganese content of infant formulas Mitchell E.J., Frisbie S.H., Roudeau S., Carmona A., Ortega R. (2020), Journal of Trace Elements in Medicine and Biology, 126710.
Line 102 « ratio of birth weight in grams to length in centimeters »
More precisely, mass divided by height cubed
Line 158 : « …age of 1170 (99.2%) infants were over 37 weeks
were -> was
Line 174 « … each 1 unit increase in ln-transformed Mn concentrations in cord blood was associated with 0.065 g/cm3 (95% CI: 0.021, 0.109) increase PI of newborns at birth (p=0.004). »
As explained above, reference values of PI at birth (mean, median, sd) are missing here to quantitatively « see » the association between manganese concentration and PI. I think a clearer indication of this relationship should added, for example that an increase in Mn concentration of 2.72 µg/L (e1) is associated with an increase in PI of 0.065 g/cm3, or about 23% of the mean PI (I made this calculation on the basis of the PI at birth of 0.283 g/cm3 but the authors' values should be used - which I have not found). This kind of comparison would help to understand if the effect of Mn is strong or weak.
Moreover, one would expect a scatter plot of PI against Mn concentration or Ln(Mn concentration). From my point of view, this graph with its linear fit should be added to the manuscript, it would give a clear and direct view of the (linear) relationship between Mn exposure and ponderal index. If possible, it should be done for all the variables studied (birth weight, height, head circumference).
Line 181. « In sex stratified analyses, higher Mn exposure was in associations with increases in PI among girls (β=0.079, 95% CI: 0.011, 0.146; p=0.022) »
The association between Mn exposure and increased PI is seen only for girls, this difference should be more discussed in the manuscript.
Line 190. Reference to supplementary table missing ?
Table S4 in the supplementary is related to paragraph 3.3 but is not cited in, the manuscript
Line 214 to 217 . « Overall, effect estimates did not change in complete analyses adding other heavy metal, such as lead and cadmium (Table S5)….. »There seems to be an inversion between Table S5 and Table S6. In the supplementary files, Table S5 refers to the adjustment for low birth weight and prematurity and Table S6 to the adjustment for Pb and Cd. Please correct.
Line 247. « ….Mn were associated with a 0.067-g increase in birth weight… »
The unit of 0.067 should be clarified. I am not sure I understand what this value is. It appears to be the β value shown in column 1 of Table 2. In this case, β is the regression coefficient of the linear model to describe the relationship between birth weight and Ln(Mn concentration) and represents the change in birth weight for each one unit change in Ln(Mn concentration). In other words, if 0.067 is the regression coefficient β, it does not represent the absolute difference in birth weight between quantiles but the slope of the linear relationship between birth weight and Ln(Mn concentration), that is very different. Sorry if I missed something but this point is not very clear to me and may also be confusing to readers.
General comment on the discussion section.
From my point of view, a striking result is that the effect of prenatal Mn exposure is sex-dependent (significant only for girls) and seems to be opposite at birth (increased weight index) compared to age 1-3 (decreased BMI). It would be useful to add some words on the possible mechanisms or hypotheses underlying the transition to a reversal of the effect.
Author Response
Dear reviewer1,
We would like to thank you for reviewing and specific editorial comments on our manuscript entitled “Cord blood manganese concentrations in relation to birth outcomes and childhood physical growth: a prospective birth cohort study” (ID: nutrients-1439092). Your thoughtful review and suggestions are valuable for us in improving the quality of the manuscript. According to the suggestions and/or comments, we have carefully revised the original manuscript and amended the relevant contents. We provide our point-by-point responses to your comments and suggestions.
Reviewer 1:
Reviewer’s brief summary:
Dai et al present a prospective epidemiological study conducted over 8 years to better understand the effect of in utero Mn exposure on growth and during childhood. This work is very interesting because the study is well done and involves a large number of children, more than 1100. As the authors point out, manganese is an essential element but is toxic in excess and is associated with negative effects on growth and neurodevelopment. Although there is a consensus on the toxicity of manganese, these effects remain difficult to measure as shown by the sometimes contradictory results of epidemiological studies. The results of Dai et al. are therefore important because they provide new information to better understand these effects.
Reply: Thank you for your reviewing and encouragement.
Please find my comments and suggestion on the manuscript:
Line 24 : « 2.9.25µg/L. »
What is the correct value ? (see 29.3 μg/L line 169)
Reply: We apologized for the mistake. We have corrected the expression in the abstract (Page 1, Line 24) and the results (Page 6, Line 174-175). The correct on is shown in the following Table R4.
Table R4. Manganese concentrations (mg/L) in umbilical cord blood. (n=1179)
|
GM |
GSD |
Min |
P25 |
P50 |
P75 |
P95 |
Max |
P |
Total |
29.03 |
1.50 |
6.84 |
22.47 |
29.25 |
37.38 |
54.55 |
316.73 |
|
Boys |
30.27 |
1.51 |
6.84 |
23.42 |
30.12 |
38.29 |
55.37 |
316.73 |
<0.001 |
Girls |
29.03 |
1.47 |
7.64 |
21.49 |
27.93 |
35.38 |
53.80 |
78.42 |
|
Abbreviations: GM: geometric mean; GSD: geometric standard deviation; Min: minimum; Max: maximum; The difference in manganese concentrations was tested by Mann-Whitney U test.
Line 25 : « β=0.065. »
At first reading, I thought that the β was a correlation coefficient (Pearson or Spearman) but it is not, it’s the regression coefficient of the linear relationship between Ln(Mn concentration) and ponderal index (if I understood correctly). This should be clearly stated in the abstract (idem for all β values).
Reply: Thank you for your pointing out. We have added the states about β (regression coefficient) and CI (confidence interval) in the abstract section.
Line 25, 26 : « Mn exposure was negatively associated with body mass index z score of children aged 1, 2, and 3 years (β=-0.375~-0.244, p<0.05) was observed… »
Two verbs, please make corrections. What means « ~ » ? is it the range of β values ? in this case replace by « to ».
Reply: Thank You! We have modified the expression in the abstract.
Line 42 : « through diet »
Suggestion « through diet and drinking water »
Reply: Thank You! We have revised.
Line 46. This is a hot topic with a world public health dimension. It would be interesting to add in the manuscript that the World Health Organization (WHO) has recently published proposed revision to its drinking water guideline for manganese. The revision is designed specifically to protect infants from excess manganese in infant formula and drinking water. (https://www.who.int/docs/default-source/wash-documents/wash-chemicals/gdwq-manganese-background-document-for-public-review.pdf). A recent paper should be added here in reference:
How much manganese is safe for infants? A review of the scientific basis of intake guidelines and regulations relevant to the manganese content of infant formulas Mitchell E.J., Frisbie S.H., Roudeau S., Carmona A., Ortega R. (2020), Journal of Trace Elements in Medicine and Biology, 126710.
Reply: Thank You! We have read WHO’s drinking water guideline for manganese and the review article. Yes, the infants might expose excess Mn from drinking water and infant formula. We added the review paper in reference when talking about the source of Mn exposure.
Line 102 « ratio of birth weight in grams to length in centimeters »
More precisely, mass divided by height cubed
Reply: Thank You! We have modified the expression in method section.
Line 158 : « …age of 1170 (99.2%) infants were over 37 weeks
were -> was
Reply: We apologized for the mistake. We have revised.
Line 174 « … each 1 unit increase in ln-transformed Mn concentrations in cord blood was associated with 0.065 g/cm3 (95% CI: 0.021, 0.109) increase PI of newborns at birth (p=0.004). »
As explained above, reference values of PI at birth (mean, median, sd) are missing here to quantitatively « see » the association between manganese concentration and PI. I think a clearer indication of this relationship should added, for example that an increase in Mn concentration of 2.72 µg/L (e1) is associated with an increase in PI of 0.065 g/cm3, or about 23% of the mean PI (I made this calculation on the basis of the PI at birth of 0.283 g/cm3 but the authors' values should be used - which I have not found). This kind of comparison would help to understand if the effect of Mn is strong or weak.
Moreover, one would expect a scatter plot of PI against Mn concentration or Ln(Mn concentration). From my point of view, this graph with its linear fit should be added to the manuscript, it would give a clear and direct view of the (linear) relationship between Mn exposure and ponderal index. If possible, it should be done for all the variables studied (birth weight, height, head circumference).
Reply: Thank you for your suggestions. As shown in Table R5, we observed that mean and standard deviation of PI were 2.59 and 0.32. In order to normalize the skewed distributions of cord blood Mn concentrations, we did the natural logarithmically transformation. Thus, a 2.92 fold increase in Mn concentration was associated with an increase in PI of 0.065, about 2.51% of the mean PI (mean: 2.59).
Following your suggestions, we re-ran the Generalized additive models. A scatter plot with linear fit (Fig 1, in the revised manuscript) about associations of Mn concentrations with birth weight and ponderal index was added in the revised manuscript. Obviously, we observed the linear relationship between prenatal Mn exposure and birth outcomes (birth weight and ponderal index) (Figure R1).
Table R5. Indicators of birth outcomes among 1179 infants.
|
Mean ± SD |
Birth weight (g) |
3491.74 ± 428.75 |
Birth length (cm) |
51.28 ± 2.29 |
Birth head circumference (cm) |
34.62 ± 1.47 |
Ponderal index (g/cm3×100) |
2.59 ± 0.32 |
Figure R1. Generalized additive models for associations between prenatal Mn exposure and birth outcomes.
Line 181. « In sex stratified analyses, higher Mn exposure was in associations with increases in PI among girls (β=0.079, 95% CI: 0.011, 0.146; p=0.022) »
The association between Mn exposure and increased PI is seen only for girls, this difference should be more discussed in the manuscript.
Reply: Thank you for your suggestion. We have added the discussion about the difference in the manuscript (Line 275-279). The sex-stratified results in PI were similar with that of the study in Bangladesh (Lee and Eum et al., 2021). The potential mechanisms of the difference might be prenatal sex-steroid hormones and maternal iron deficiency (Rivera-Núñez and Ashrap et al., 2021; Kupsco and Estrada-Gutierrez et al., 2020).
Reference:
Lee, M. and K. Eum, et al. (2021). "Umbilical Cord Blood Metal Mixtures and Birth Size in Bangladeshi Children." Environmental Health Perspectives 129 (5).
Rivera-Núñez, Z. and P. Ashrap, et al. (2021). "Association of biomarkers of exposure to metals and metalloids with maternal hormones in pregnant women from Puerto Rico." Environment International 147: 106310.Kupsco A, Estrada-Gutierrez G,
Kupsco, A. and G. Estrada-Gutierrez, et al. (2020). "Modification of the effects of prenatal manganese exposure on child neurodevelopment by maternal anemia and iron deficiency." Pediatr Res 88 (2): 325-333.
Line 190. Reference to supplementary table missing ?
Table S4 in the supplementary is related to paragraph 3.3 but is not cited in, the manuscript
Reply: Thank you for your suggestion. We have cited the supplementary table S4 in paragraph 3.3.
Line 214 to 217 . « Overall, effect estimates did not change in complete analyses adding other heavy metal, such as lead and cadmium (Table S5)….. »There seems to be an inversion between Table S5 and Table S6. In the supplementary files, Table S5 refers to the adjustment for low birth weight and prematurity and Table S6 to the adjustment for Pb and Cd. Please correct.
Reply: Thank you for your suggestion. We have corrected the table serial number.
Line 247. « ….Mn were associated with a 0.067-g increase in birth weight… »
The unit of 0.067 should be clarified. I am not sure I understand what this value is. It appears to be the β value shown in column 1 of Table 2. In this case, β is the regression coefficient of the linear model to describe the relationship between birth weight and Ln(Mn concentration) and represents the change in birth weight for each one unit change in Ln(Mn concentration). In other words, if 0.067 is the regression coefficient β, it does not represent the absolute difference in birth weight between quantiles but the slope of the linear relationship between birth weight and Ln (Mn concentration), that is very different. Sorry if I missed something but this point is not very clear to me and may also be confusing to readers.
Reply: We are sorry for this confusion! To avoid confusion, we have corrected the expression in the discussion section. As your thought, when exploring associations between the quantiles of Mn concentrations and birth weight, the variable of quantiles was added in the models as categorial variable. Additionally, we consider the lowest levels of Mn concentrations as the reference.
General comment on the discussion section.
From my point of view, a striking result is that the effect of prenatal Mn exposure is sex-dependent (significant only for girls) and seems to be opposite at birth (increased weight index) compared to age 1-3 (decreased BMI). It would be useful to add some words on the possible mechanisms or hypotheses underlying the transition to a reversal of the effect.
Reply: We appreciated your thoughtful comments. Following your suggestions, we have added some possible mechanisms or hypotheses about the striking results in the discussion section (Line 270-279, Line 300-306).
Best regards,
Sincerely Yours,
Zhijun Zhou
Reviewer 2 Report
How does EDTA affect the availability of Manganese for analysis?
what is the quality of the acid used?
How do you ensure the absence of environmental contamination in the materials used?
the discussion must be improved to be able to explain the results with scientific solidity
Author Response
Dear reviewer,
We would like to thank you for reviewing and specific editorial comments on our manuscript entitled “Cord blood manganese concentrations in relation to birth outcomes and childhood physical growth: a prospective birth cohort study” (ID: nutrients-1439092). Your thoughtful review and suggestions are valuable for us in improving the quality of the manuscript. According to the suggestions and/or comments, we have carefully revised the original manuscript and amended the relevant contents. We provide our point-by-point responses to your comments and suggestions.
Reviewer 2:
Comments and Suggestions for Authors
How does EDTA affect the availability of Manganese for analysis?
Reply: Thank you for your careful review. Yes, Mn could be chelated by amino ligands such as ethylenediaminetetraacetic acid (EDTA) and Mn chelates are less soluble in alkaline conditions. In our method of cord blood Mn analysis, we diluted samples with 0.1% nitric acid to dissolve Mn, and chose 500~550℃ as the ashing temperature and 2500℃ as atomization temperature. Thus, impacts from EDTA on Mn analysis could be effectively reduced.
Reference
Gao Y, Zhou Y, Pang SY, et al. Enhanced peroxymonosulfate activation via complexed Mn(II): A novel non-radical oxidation mechanism involving manganese intermediates. Water Res. 2021;193:116856. doi:10.1016/j.watres.2021.116856
Li M, Hu L, Zhong H, He Z, Sun W, Xiong D. Efficient removal of diethyl dithiocarbamate with EDTA functionalized electrolytic manganese residue and mechanism exploration. J Hazard Mater. 2021;410:124582. doi:10.1016/j.jhazmat.2020.124582
what is the quality of the acid used?
Reply: Thank you for your careful review. The quality of the acid used in analysis was Guaranteed reagent (GR).
How do you ensure the absence of environmental contamination in the materials used?
Reply: Thank you for your careful review. To avoid the contamination, blood samples collection and Mn analysis in the laboratory were done by following the standard protocols. Umbilical cord blood samples were collected by professional midwives. Whole blood samples were collected in 5ml sterile centrifuge tubes containing anticoagulant EDTA and stored at -80℃ until analysis. The tubes used for dilution were the high-density polypropylene centrifuge tubes (Corning Incorporated, USA). Reagent was free from metal contamination.
the discussion must be improved to be able to explain the results with scientific solidity
Reply: We quite appreciate your valuable comments. Following the advice of reviewers, we revised the results and discussion, described the results more clearly and explained more about results of sex difference, and the inverse results at birth compared with that at age of 1-3 years.
Best regards,
Sincerely Yours,
Zhijun Zhou
Round 2
Reviewer 2 Report
It is advisable to include purity of reagents and cleaning methods of the material that causes the absence of contamination.
Author Response
Dear reviewer,
We would like to thank you for reviewing and specific editorial comments on our manuscript entitled “Cord blood manganese concentrations in relation to birth outcomes and childhood physical growth: a prospective birth cohort study” (ID: nutrients-1439092). Your thoughtful review and suggestions are valuable for us in improving the quality of the manuscript. According to the suggestions and/or comments, we have carefully revised the original manuscript and amended the relevant contents. We provide our point-by-point responses to your comments and suggestions.
It is advisable to include purity of reagents and cleaning methods of the material that causes the absence of contamination.
Reply: Thank you for your suggestions. The quality of the acid used in analysis was Guaranteed reagent (GR). We have added the purity of reagents in the methods section. The tubes used for dilution were the high-density polypropylene centrifuge tubes (Corning Incorporated, USA). Reagents and materials were free from metal contamination.
Best regards,
Sincerely Yours,
Zhijun Zhou